# Searching for the Optimal Donor for Allogenic Adipose-Derived Stem Cells: A Comprehensive Review

**DOI:** 10.3390/pharmaceutics14112338

**Published:** 2022-10-29

**Authors:** Tihomir Georgiev-Hristov, Mariano García-Arranz, Jacobo Trébol-López, Paula Barba-Recreo, Damián García-Olmo

**Affiliations:** 1Servicio de Cirugía General y del Aparato Digestivo, Hospital General Universitario de Villalba, 28400 Madrid, Spain; 2Facultad de Medicina, Universidad Alfonso X, 28691 Madrid, Spain; 3Instituto de Investigación Sanitaria, Hospital Universitario Fundación Jiménez Díaz, 28040 Madrid, Spain; 4Departamento de Cirugía, Universidad Autónoma de Madrid, 28029 Madrid, Spain; 5Servicio de Cirugía General y del Aparato Digestivo, Complejo Asistencial Universitario de Salamanca, 37007 Salamanca, Spain; 6Servicio de Cirugía Maxilofacial, Hospital Universitario Rey Juan Carlos, 28933 Madrid, Spain

**Keywords:** adipose-derived stem cells, stem cell donor, allogenic stem cells

## Abstract

Adipose-derived stem cells comprise several clinically beneficial qualities that have been explored in basic research and have motivated several clinical studies with promising results. After being approved in the European Union, UK, Switzerland, Israel, and Japan, allogeneic adipose-derived stem cells (darvadstrocel) have been recently granted a regenerative medicine advanced therapy (RMAT) designation by US FDA for complex perianal fistulas in adults with Crohn’s disease. This huge scientific step is likely to impact the future spread of the indications of allogeneic adipose-derived stem cell applications. The current knowledge on adipose stem cell harvest describes quantitative and qualitative differences that could be influenced by different donor conditions and donor sites. In this comprehensive review, we summarize the current knowledge on the topic and propose donor profiles that could provide the optimal initial quality of this living drug, as a starting point for further applications and studies in different pathological conditions.

## 1. Introduction

Adipose tissue is a well-known source of adipose-derived stem cells (ADSCs). Since 2003, ADSCs have gained a growing importance due to their properties, mainly immunomodulation, differentiation capacity, and proangiogenesis. Since it was found that ADSCs lack MHC class II antigen (HLA-DR) and express only low levels of MHC class I antigens (HLA-A,B,C), they were considered immune-evasive as they could be allogenously transplanted without major adverse events [1,2]. Currently, no advantages of autologous vs. allogenous transplantation of ADSCs could be demonstrated [3]. As significant evidence for the dose-dependent effect continuously accumulates, scientific progress in the field has moved towards the establishment of stem cell banks that could provide cells for clinical studies, compassionate use, and recently for approved clinical indication (Crohn’s disease complex perianal fistula). The application of both autologous and allogeneic ADSCs has proven to be safe, but the use of allogeneic ADSCs has the advantages of availability (time), quality, cell homogeneity, and cost (Table 1).

The results of the continuously increasing number of clinical studies finally lead to a recognition of allogeneic ADSCs (darvadstrocel) as the first living drug by the European Medicines Agency (EMA) in 2018, approved for the treatment of Crohn’s disease related anal fistulous pathology in adults. The same approval was also given in the UK, Switzerland, Israel, and Japan, and recently a RMAT (regenerative medicine advanced therapy) designation was granted to darvadstrocel by the US FDA. This huge scientific step is likely to impact the future spread of the indications, exploring new conditions in controlled clinical trials that would eventually lead to further approved therapies for allogeneic ADSC applications. However, for indications other than perianal complex Crohn’s related fistulae, ADSCs are still within the experimental stage, and a great variety of ADSCs cellular products are being used that further complicate the extrapolation of the results. This situation raises the question of whether all types of ADSCs are the same, and which are the factors that could influence the cell quantity and quality and could further modify the results.

We could identify several crucial levels where different conditions could influence the stem cell characteristics and, eventually, the expected clinical results:
Donor-to-recipient critical steps
HarvestingProcessing
○Initial manipulation○Culture and expansion○Storage○Package and transportation
Handling and application
Donor factors
Donor characteristicsDonor tissue type and site


It is known that small modifications of processes of harvesting, processing, handling, and application of ADSCs could critically impact the expected results of the cell therapy and therefore could be subjects of separated papers. These are not addressed in the present study.

## 2. Donor-to-Recipient Critical Steps

To be implanted in a patient, ADSCs pass throughout several critical steps that could modify their properties. Although this issue has not been previously studied in an organized and controlled fashion, important knowledge has been gathered from the experience in clinical trials and preclinical studies. Some of the conclusions of that research have been introduced as requirements for their use by official organizations, such as EMA or FDA. National and international governmental organizations have created legal requirements for cell and tissue transplantation, tissue engineering procedures, and regenerative medicine application that regulate the issues from donor selection, minimally required laboratory examinations, and permissions to be obtained based mainly on the safety of the cell/tissue manipulation throughout all the steps [4]. For example, the legislation (European Directive 2004/23/EC and American 21 CFR 1271) demands minimum requirements for donor selection that guarantee the traceability of the donated cells and their safety. In this respect, a donor must demonstrate the absence of infectious diseases (HBV, HCV, VIH, SARS-CoV-2, Zika virus, WNV, CMV, parvovirus B19, *Treponema pallidum..*.). These good manufacturing practices (GMPs) are continuously updated and synchronized between the different national and international agencies, aiming to create a safe environment for scientific progress and minimize the effect of ADSC processing on their properties [5]. Currently, the EMA and FDA regulations specific to advanced therapies products (including allogenic ADSCs) outline all the main issues of non-conventional drug manufacturing supported by the risk-based approach (Table 2). Specifically, these include qualification of personnel, as well as qualification and validation of facilities, equipment, documentation, raw materials and excipients, aseptic production, testing methods, and quality control (at least composition, viability, cell potency, and microbiological safety), batch release, and distribution. However, other factors could be identified, and their effect on the cell quantity and function, which could benefit or humper the clinical results, has not been discussed in the legislation framework and deserves profound evaluation. A recently published review attempts to provide guidance for better and homogeneous manufacturing of therapeutic cellular products with special reference to MSCs [6].

The current research on adipose stem cell harvest describes quantitative and qualitative differences that could be influenced by different donor conditions and donor sites. In this comprehensive review, we sought to summarize the current knowledge on the topic and propose donor profiles that could provide the optimal initial quality of this living drug as a starting point for further applications and studies.

## 3. Donor Factors

### 3.1. Age

Ageing is known to have a negative impact on all the human tissues and cells, including stem cells. ASCs aging has been demonstrated by differential expression of miRNA in younger (<35 years-old) and older (>60 years-old) donors, and this translated into reduced regeneration capacity [7]. As most of the functions expressed by the ADSCs are cytokine-mediated, a possible alteration of the secretome could lead to further functional changes. It was found that secretory profile of ADSCs is altered in aged donors, with reduced secretion of VEGF, HGF, and SDF-1α, and increased TGF-β production. These findings could further explain the reduced immunomodulatory and angiogenic capacities found in ADSCs from aged donors [8,9,10]. ADSCs are found to express a senescence-associated profile that includes *β*-galactosidase activity, enlarged morphology, and p53 protein upregulation that could explain the decreased proliferation capacity observed in culture media [11,12,13]. However, ageing does not affect equally all ADSC properties, and some contradictory data have been published in the literature. Girolamo et al. showed that cell viability and in vitro adipocytic differentiation were not significantly affected by ageing, whereas osteoblastic differentiation capacity was hampered [14]. On the contrary, other authors did not find any significant donor age-related differences of the osteogenic properties [15,16]. In recent years, numerous studies have been conducted that analyzed the effect of the age of ADSC donors. In 2013, Wu et al. compared cells from infants, adults, and elderly, and demonstrated a loss of viability and regenerative potential associated with increasing donor age [16]. Similar results have been obtained by Zhang et al. in 2018 and Park et al. in 2022 [17,18].

### 3.2. Gender

Although earlier studies failed to prove any significant yield and functional differences between male and female ADSCs, more recent research has unveiled this issue by more sophisticated bioinformatic tools, analyzing the molecular and genetic dimorphism that could drive gender-related ADSC differences. Bianconiet al. recently performed a systematic meta-analysis of hMSC microarrays using the Transcriptome Mapper (TRAM) software [19]. They identified several chromosomal segments and differentially expressed genes in male and female ADSCs related to inflammation, differentiation capacity, and paracrine mechanisms. These findings could be further demonstrated mainly in vitro in other studies, strengthening the conclusion of the gender influence on the ADSC functionality. It was found that female ADSCs have a higher immunosuppression capacity compared to male ADSCs, coordinated by increased levels of anti-inflammatory cytokines IDO1, IL-1RA, and PGE-2, and lower levels of pro-inflammatory cytokines such as G-CSF [20]. The authors found that female (but not male) ADSCs downregulated IL-2 receptor and induced a sustained expression of CD69 in peripheral blood mononuclear cells. On the other hand, their results suggest no need for gender matching, as the immunosuppressive effect of ADSCs remained stable after female-derived ADSCs were co-cultured with peripheral blood mononuclear cells of both sexes. Ogawa et al. found in an in vitro study that ADSCs from female donors have higher adipogenic differentiation capacity than male-derived ADSCs [21]. Gender was also identified to be an important factor that impacts the paracrine, differentiation, and proliferation capacity. In their study, Shu et al. found that ADSCs from female donors exhibit a better ability to differentiate towards bone, fat, and muscle tissue and higher secretion capacity of VEGF and HGF, with a lower apoptotic rate [22]. Although it seems that ADSCs from female donors could be functionally superior, in some studies, male ADSCs, especially from superficial fat tissue, obtained from abdominoplasty specimens proved to be more efficient in achieving osteogenesis [23].

### 3.3. Immune Conditions

Having immunomodulatory activity, it seems logical that ADSC’s functions could be influenced by certain immune diseases. Crohn’s disease is currently one of the main target diseases for stem cell application. However, it has been found in previous studies that autologous ADSCs are less effective in the treatment of perianal fistulae compared to the allogenic ADSCs. Although ADSC yield from inflammatory bowel disease patients was higher [24], an in vitro study of mesenteric and subcutaneous fat tissue from Crohn’s disease patients and healthy donors found that Crohn’s disease patients’ ADSCs expressed more proinflammatory (IL6, TNFA, CCL2, and IL1B), invasive, and phagocytic phenotype and reduced immunosuppressive properties [25]. Similarly, ADSCs derived from ulcerative colitis patients express an altered immunosuppressive profile consisting of lower prostaglandin E2, idoleamine 2, 3-dioxygenase, and TNF-alfa-induced protein 6 [26]. These findings suggest that ADSCs from donors with immune conditions may not be appropriate due to their deficiency in terms of immunomodulatory capacity.

### 3.4. Diabetes

Donor metabolic conditions could also alter the immunomodulatory activity of the ADSCs. Serena et al. found that obesity and Type 2 Diabetes promote the expression of a proinflammatory profile by the ADSCs [27]. Furthermore, Diabetes Mellitus hampers the secretory (through reduced secretion of VEGF, adiponectin, and CXCL-12) and proliferative activity, exhibiting mitochondrial disfunction and senescence phenotype [28]. These findings suggest that ADSCs from diabetic donors should be avoided as their initial characteristics predict altered functionality. However, it seems that ADSCs from different sites are also different in their characteristics. Therefore, not surprisingly, ADSCs from peripancreatic fat tissue of diabetic patients were found to maintain the migration, immunomodulatory, chondrogenic differentiation capacities, stemness, and vitality as in non-diabetic subjects, while only adipogenic and osteogenic capacity were altered [29]. Osteogenic capacity of ADSCs from diabetic patients is a point of controversy, as other studies have suggested even increased osteogenic potential based on the mRNA level of *BGLAP, ALP,* and *SPP1* [30].

### 3.5. Obesity

Obesity is a well-known proinflammatory state [31]. Although some studies have not found differences in the ADSC yields and proliferation capacity [32,33], more recent studies, based on gene expression, have found important alterations. The altered microenvironment in morbidly obese patients, characterized by increased levels of pro-inflammatory cytokines, is found to impact the ADSC functionality [34]. Roldan et al. described a general short-circuit of the stemness gene network of ADSCs from obese donors [35]. Oñate et al. found that ADSCs from morbidly obese patients have a lower proliferation, differentiation, and proangiogenic capacity, as demonstrated by higher TSP-1 and VEGFR1 expression [36]. Although obesity is considered a factor that decreases the immunomodulation capacity of ADSCs [37], in a study of weight-discordant monozygotic twins, it was found that higher weight is related to a lower angiogenic capacity of the ADSCs, but the immunomodulatory activity was stronger, as well as the adipogenic differentiation capacity [38]. Furthermore, ADSCs from obese donors are found to induce an in vitro proinflammatory profile in murine macrophages and microglial cells [39]. ADSCs from obese donors (age and sex matched) produce smaller extracellular vesicles than lean ADSCs, with dysregulation of their miRNA cargo, which alters the cell capacity to modulate injury pathways [40]. These functional alterations caused by obesity seem to be donor site-dependent, as described in the paper of de Girolamo et al., where they found a higher degree of functional and stemness impairment within the visceral fat of obese patient [41]. The presence of metabolic syndrome in those patients could further worsen the ADSC osteogenic and proliferation capacity, which were generally found in obese patients [42,43].

### 3.6. Lifestyle Habits

An increasing number of studies are linking different lifestyle habits to the quantity and quality of ADSCs obtained from liposuction. For example, the use of e-cigarettes [44] and tobacco by-products, such as nicotine, have been shown to have a detrimental effect on the obtained ADSCs and their differentiation capacities [45,46,47]. Another example is that regular alcohol consumption induces a lower potential, as well as a decrease in the number of mesenchymal stromal cells [48,49,50].

### 3.7. Donor Site

Multiple studies have addressed the search for an optimal donor site to obtain the highest quantity and functionality of ADSCs. Studies oriented towards obtaining of fat grafts for the plastic and esthetic procedure purposes mainly inform on the cellularity and viability, and only some papers study the differentiation capacity. The lower abdomen and inner thigh seem to yield higher cellularity with greater viability of the cells obtained from the upper abdomen [23,51,52,53], although the outer thigh has also been found to be favorable [54]. This fact itself would not necessarily translate into improved functionality. In fact, Jurgens et al. did not find any osteogenic differentiation capacity differences between different sites [53]. Other studies have found that ADSCs from flanks and thighs express an increased osteogenic and decreased adipogenic capacity compared to ADSCs from the abdomen [55]. ADSCs obtained from thigh subcutaneous fat were also found to have an increased angiogenic potential (higher VEGF, VEGF2, and CD31 expression) compared to abdominal fat tissue [56]. In the same study, the authors describe an increased adipogenic capacity in the thigh-derived ADSCs compared to the abdominal-derived ADSCs, in disagreement with findings from the paper cited above. Similar superior results were found with ADSCs from the gluteal fat tissue [57]. Within the abdominal subcutaneous tissue, it seems that superficial fat (above Scarpa’s fascia) could have higher yield and adipogenic capacity, as well as increased multipotency and stemness [58,59]. Other possible sources of ADSCs have also been explored. Omental, percicardial, mediastinal, synovial, and other specific localizations of fat tissue have been studied in a limited number of studies, and their characteristics seem favorable for treatment purposes of inflammatory, regenerative, or ischemic issues of nearly located organs [29,60,61]. Although ADSCs from different sites express the same surface markers, they are proven to be genetically different and express different capacities. For example, epicardial and omental ADSCs were found to have a higher osteogenic and adipogenic potential than pericardial ADSCs, but only the epicardial ADSCs exhibit a high cardyomyogenic potential [61,62]. However subcutaneous ADSCs have higher proliferation and adipogenic capacity than visceral ADSCs [62,63,64].

## 4. Discussion

Although the legislation does not differentiate between autologous and allogeneic treatments in advanced therapies, we have considered to delve into what we consider to be the critical aspects, based on our experience, of healthy donors for allogeneic ADSC applications. Many aspects that we have analyzed are difficult to propose for autologous use. As the patient is the donor, it is difficult to screen for age, gender, weight, immune conditions, and even lifestyle habits of the patients. Nevertheless, all the described above recommendations could be considered valuable in autologous use to obtain better results.

Another important aspect is the high cost of producing and treating advanced therapies. In this sense, allogeneic use would have some advantages, such as limiting the risk of contamination during the production process, homogenization of the obtained cell product (greater potency in selected donors), and lower production cost (cells for several treatments can be obtained from one donor).

### 4.1. Donor Selection

The number of stem cell studies during the last two decades has increased exponentially. However, many of the indications still have not surpassed the current limit of experimental indication towards routine clinical use. The main reason for the lack of expected progress is the great variability of the results from the phase 3 clinical studies. Taking into consideration the above explained numerous factors that could influence the autologous use and the lack of standardization on the donor profile in allogenous applications, together with the lack of standardization on some of the technical aspects of the whole process of stem cell treatment, from isolation to applications, we could conclude that the results could be highly biased and this could impede obtaining conclusions on the real benefits of ADSC application in the studied pathological conditions.

Without being explicitly determined to establish strict guidelines for every single step of application, from the stem cell harvesting to the application on the receptor, the scientific progress of the field has uncovered basic principles for a successful application, establishing the optimal conditions to obtain maximum results. Furthermore, the functionality of ADSCs derived from elderly, diseased, and obese patients was found to be altered and therefore they cannot be considered an optimal cell therapy source, turning the allogeneic cells into a more suitable option for clinical application in these cases. The altered proliferation capacity of aged donor-derived cells could further complicate the process of obtaining a high quantity of cells that are needed for most of the indications. Recent and promising studies have tried to better understand the lack of clear beneficial clinical effects in otherwise well-designed clinical trials after promising preclinical and early clinical results, looking deeper into the molecular and gene-expression profile of ADSCs from responders and non-responders [65]. The authors found that an ADSC profile of lower proliferation rate, lower proinflammatory molecule secretion, and higher osteoblast differentiation capacity of the implanted ADSCs was associated with better perianal fistula healing in the treated patients. This study brings some light to the questions of the heterogenous results in autologous ADSC clinical trials and advocates for a standardization of the donor profiles. We believe that donor standardization could be more successful and detailed in allogenic transplantation provided from ADSC banks. This process could include legislation and regulation issues but also warrants scientific publication requirements to facilitate further comparison of the published literature in the future and could eventually explain the highly variable results between different studies.

In order to minimize the impact of individual donor characteristics, Widholz et al. described a pooling method to cultivate together ADSCs from several donors without hampering their characteristics [66]. However, this could be only a partial solution if a previous donor profiling and individual donor ADSC characterization has not been properly undertaken. Taking into consideration all the possible factors that could hamper the functionality of ADSCs, we believe that autologous application in the future could have a very limited use and would shift towards allogenic use in order to better control the quality of the cells, achieving high quantities and selecting optimal donors that fulfill a number of requisites and preliminary tests, eliminating the innate problems, manifested or not, that could hamper the autologous stem cell functions. This strategy could allow the creation of specific ADSC profiles, which could be used according to the pathological condition to be treated.

### 4.2. Preconditioning

Stem cells, as a functional unit of the human body, comprise properties that are being continuously unveiled to benefit the medicine. At the time of harvesting from the donor tissue, stem cells possess a certain level of homeostasis that is influenced by numerous individual conditions that interact with the cell functions and could possibly alter their capacity to respond to certain stimuli. Some of the factors are already well studied (age, health status, donor site). However, many other factors could influence the properties of the obtained cells. Even if we select the perfect donor, many other questions remain to be answered. It is still unknown whether cultured stem cells maintain the same functional characteristics in vivo as the initially harvested ADSCs, as well as the fate of the implanted cells from a functional point of view. It is well known that implanted cells have a very low rate of engraftment. It could be partially explained by the unfavorable local conditions of the disease to be treated (inflammation, infection, ischemia, etc.) and by the sudden change of the local conditions for the implanted cells (from a culture with standard level of nutrients, O_2_ and CO_2_, etc., to the hypoxic tissue or cavity) and could further worsen the expected results. In order to overcome this issue, currently, the most common strategy relies on the increasing the number of the implanted cells. Other possible options are based on the improving the initial properties of the obtained ADSCs [67]. However, most of these strategies, based on genetic modification or pre-activation, are still on a preclinical level of research, and soon we must consider only those that do not significantly alter the legally allowed production protocols due to safety reasons. For example, hypoxic preconditioning is found to improve the angiogenetic properties of ADSCs, and this could partially restore this altered function of ADSCs from elderly donors by up-regulation of proangiogenic factors (VEGF, PlGF, and HGF) and down-regulation of antiangiogenic factors (TBS1 and PAI-1) [9]. Hypoxia has multiple beneficial effects on ADSCs. Hypoxic environment itself is not an exact term, as oxygen concentration in the subcutaneous fat tissue is much lower than in the air (1% vs. 21%, respectively). Therefore, efforts should be made in the future to achieve culture conditions, similar to those of tissue of origin, including 3D culture systems, for example.

ADSC preconditioning could also be possible even before the cell harvesting, mainly by metabolic changes of the donor’s homeostasis. It was found that the altered properties of ADSCs from obese patients could be partially reprogrammed after weight loss (after bariatric surgery or long-term diet) [68]. Other ways for cell reprogramming in order to enhance certain ADSC functions in vitro have been studied. However, the currently available legislation limits the possibility of clinical applications of this strategy (for example genetic modification, preconditioning with growth factors of mechanical stimuli). Based on the above results, analyzing the ideal ADSC donor characteristics, we could conclude that the best cell yield, in terms of quantity and quality, is obtained from a young woman (preferably under 40), without immune or inflammatory pathologies, with a BMI between 17.5 and 26, not excessively athletic (very fibrotic subcutaneous tissue), and with healthy living habits (no smoking, no regular alcohol consumption). It is also preferable to obtain the fat tissue by manual liposuction of thigh subcutaneous fat or from the gluteal fat tissue.

## 5. Limitations of the Study

The comparison of some of the reviewed studies could be biased, as these studies are mostly in vitro research papers, and cell behavior under the stress conditions of the implantations could be completely different. Furthermore, some of publications on functional ADSC studies over a single factor do not consider and control for other factors. For example, studies centered on the gender effects do not consider the BMI of the donor. Even more, fat tissue from abdominoplasty specimens is often used, as it is a discarded tissue of other surgical procedures. The fact that abdominoplasty was performed could mean that a certain degree of obesity was present in those patients, and this could further bias the conclusion draw.

Besides these limitations, the growing amount of data allows us to state that numerous patient factors could influence the ADSC initial functions. Further studies, taking all of them into account, together with the final functional analysis of ADSCs before application, could shed lighter on this issue and could reach the final goal of achieving an optimal donor.

Legislation has already placed several limits that are rather restrictive but aim to protect the valuable quality of this new treatment, discarding all the possible pathological factors that could jeopardize the results. As the scientific field of stem cell application is continuously evolving, legislation is also progressively adapting to the new data to improve the results and offer a controlled and ethically safe environment for further development.

## 6. Conclusions

There is likely no such thing as a perfect donor. However, several factors must be taken into consideration in order to begin with the most beneficial cell population, which seem to be more efficient for the individual pathological condition to be treated. As published studies are scarce, sometimes controversial, and describe only certain characteristics of the cells in a specific scenario, it could be difficult to extrapolate the recommendations to other applications.

On the other hand, it is theoretically possible that certain cell characteristics could be beneficial, and others could be deficient, depending on the donor characteristics and harvesting site (Figure 1). It means that a particular cell product could still have optimal characteristics for one indication and insufficient characteristics for another. These strategies have not been studied yet and we believe that, based on the published literature, we could optimize the donor selection, creating donor profiles and thereafter comparing the possible functional differences. After selecting the most appropriate donor profile, certain maneuvers of preconditioning could further prepare the cells for the treated pathological situation. For example, if we seek an optimal immunomodulatory activity to treat a chronic inflammation, we must choose a young, healthy, not obese, female donor, and obtain stem cells from the outer thigh, gluteal region, or lower abdomen. Next, we could consider a culture under hypoxic conditions and perform an application carefully, following the implantation protocol strictly. It is possible that the same cells have a suboptimal performance if applied for pancreatic endocrine regeneration purpose that could be better addressed by peripancreatic ADSCs.

We believe this review could open the gate for further studies that could complete the profile of an optimal donor for every condition to be treated, evaluating other possible factors (endocrine status, nutritional status, recent metabolic changes, tobacco use, etc.), together with the possibility of preconditioning the cells according to the current ethical and legislation framework.

## Figures and Tables

**Figure 1 pharmaceutics-14-02338-f001:**
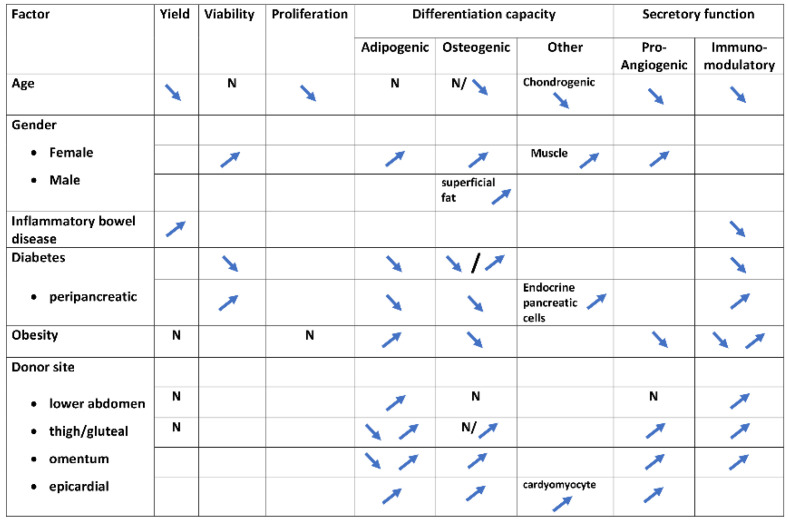
ADSC characteristic alterations by individual donor factors. Schematic summary of the reviewed literature. N—not significantly altered; Upstream arrow—potentiation of the characteristic; Downstream arrow—inhibition of the characteristic.

**Table 1 pharmaceutics-14-02338-t001:** Principal advantages of allogenic cells.

Optimal donor selection
No waiting time for treatment (multiplication process)
Availability of large number of cells
Minimizing cell contamination (in production)
Limiting the variability of ADSCs (donor pathologies)
Lower costs

**Table 2 pharmaceutics-14-02338-t002:** Rules and guidelines that regulate advanced therapies medicinal products (including allogeneic ADSCs).

Rules and Guidelines	Webpage/Link
- EU GMP-ATMP: EudraLex-The Rules Governing Medicinal Products in the European Union. Volume 4: Good Manufacturing Practice. Guidelines on Good Manufacturing Practice specific to Advanced Therapy Medicinal Products. 22 November 2017.	https://health.ec.europa.eu/system/files/2017-11/2017_11_22_guidelines_gmp_for_atmps_0.pdf (accessed on 1 September 2022)
- ICHQ5D: Quality of Biotechnological Products: Derivation and Characterization of Cell Substrates Used for Production of Biotechnological/Biological Products. CPMP/ICH/294/95. (1998).	https://www.ema.europa.eu/en/documents/scientific-guideline/ich-q-5-d-derivation-characterisation-cell-substrates-used-production-biotechnological/biological-products-step-5_en.pdf (accessed on 1 September 2022)
- CPMP/ICH/138/95: Note for guidance on quality of biotechnological products: stability testing of biotechnological/biological products.	https://www.ema.europa.eu/en/documents/scientific-guideline/ich-topic-q-5-c-quality-biotechnological-products-stability-testing-biotechnological/biological-products_en.pdf (accessed on 1 September 2022)
- CMCa: Guidance for FDA Reviewers and Sponsors: Content and Review of Chemistry, Manufacturing, and Control (CMC) Information for Human Gene Therapy Investigational New Drug Applications (INDs) (2008).	https://permanent.fdlp.gov/LPS111884/LPS111884_gtindcmc.pdf (accessed on 1 September 2022)
- CMCb: Guidance for FDA Reviewers and Sponsors: Content and Review of Chemistry, Manufacturing, and Control (CMC) Information for Human Somatic Cell Therapy Investigational New Drug Applications (INDs) (2008).	https://www.fda.gov/media/73624/download (accessed on 1 September 2022)
- FDA-2008-D-0520: Guidance for Industry: Potency Tests for Cellular and Gene Therapy Products (01/2011).	https://www.fda.gov/media/79856/download (accessed on 1 September 2022)

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
