# Peer review of "Searching for the Optimal Donor for Allogenic Adipose-Derived Stem Cells: A Comprehensive Review"

_pharmaceutics, 2022, doi:10.3390/pharmaceutics14112338_

Round 1

Reviewer 1 Report

The present MS is devoted to the important issue in the field of regenerative medicine – great variability of cellular products from different donors. The main stream of this review is the analysis of some individual features of donors whose adipose tissue is used to prepare the cell preparations.

The authors do not discuss in detail the clinical results of different steps of processing, expansion, preparation and storage of cell products, although Section 2 briefly mentions these issues. Unfortunately, this MS did not “open the gate for further study” because it is impossible to “complete profile for an optimal donor” for every disease even evaluating many factors. We need minimal (but strict) standardization for processing procedures and joint requirements for donors during preclinical and clinical studies. 

Reviewer 2 Report

 This work contains many strengths but nevertheless several aspects should be reviewed:

In the introduction (line 45-46), the authors comments that the allogenic ADSC has advantages of safety. Why? Autologous cells are safe too. This point should be justified.

What are the main attributes to be determined in the release of a batch of allogeneic products?

What it is mead: living drug? (Line 53), may would be better define it as ATMP

Allogeneic ADSCs (darvadstrocel) is too know how alofisel, please specific.

Several crucial levels are described for allogenic cells: Donor-to-recipient critical steps and

Donor factors,  but these point should be compared with autologous cells.

Could the authors provide a summary table with the main characteristics described in points 2 Donor-to-recipient critical steps and 3.Donor factors.

Discussions: this section should justify the advantages and disadvantages of ADSC compared to current products and therapies.

Round 2

Reviewer 1 Report

no additional comments to revised version   

Reviewer 2 Report

Some questions have not been incluide in the final version of manuscript:

The main attributes to be determined in the release of a batch of allogeneic products, this information should be include in the manuscript.

Several crucial levels are described for allogenic cells: Donor-to-recipient critical steps and Donor factors, but these point should be compared with autologous cells.

The answer should be included in the manuscritp.
